# Anti-Human CD9 Fab Fragment Antibody Blocks the Extracellular Vesicle-Mediated Increase in Malignancy of Colon Cancer Cells

**DOI:** 10.3390/cells11162474

**Published:** 2022-08-10

**Authors:** Mark F. Santos, Germana Rappa, Simona Fontana, Jana Karbanová, Feryal Aalam, Derek Tai, Zhiyin Li, Marzia Pucci, Riccardo Alessandro, Chikao Morimoto, Denis Corbeil, Aurelio Lorico

**Affiliations:** 1Department of Basic Sciences, Touro University College of Medicine, Henderson, NV 89014, USA; 2Department of Biomedicine, Neurosciences and Advanced Diagnostics, University of Palermo, 90133 Palermo, Italy; 3Tissue Engineering Laboratories, Biotechnology Center (BIOTEC) and Center for Molecular and Cellular Bioengineering, Technische Universität Dresden, 01307 Dresden, Germany; 4Institute for Biomedical Research and Innovation (IRIB), National Research Council (CNR), 90146 Palermo, Italy; 5Department of Therapy Development and Innovation for Immune Disorders and Cancers, Graduate School of Medicine, Juntendo University, Tokyo 113-8421, Japan

**Keywords:** cancer, CD9, Fab, cell morphology, migration, colon carcinoma, extracellular vesicle

## Abstract

Intercellular communication between cancer cells themselves or with healthy cells in the tumor microenvironment and/or pre-metastatic sites plays an important role in cancer progression and metastasis. In addition to ligand–receptor signaling complexes, extracellular vesicles (EVs) are emerging as novel mediators of intercellular communication both in tissue homeostasis and in diseases such as cancer. EV-mediated transfer of molecular activities impacting morphological features and cell motility from highly metastatic SW620 cells to non-metastatic SW480 cells is a good in vitro example to illustrate the increased malignancy of colorectal cancer leading to its transformation and aggressive behavior. In an attempt to intercept the intercellular communication promoted by EVs, we recently developed a monovalent Fab fragment antibody directed against human CD9 tetraspanin and showed its effectiveness in blocking the internalization of melanoma cell-derived EVs and the nuclear transfer of their cargo proteins into recipient cells. Here, we employed the SW480/SW620 model to investigate the anti-cancer potential of the anti-CD9 Fab antibody. We first demonstrated that most EVs derived from SW620 cells contain CD9, making them potential targets. We then found that the anti-CD9 Fab antibody, but not the corresponding divalent antibody, prevented internalization of EVs from SW620 cells into SW480 cells, thereby inhibiting their phenotypic transformation, i.e., the change from a mesenchymal-like morphology to a rounded amoeboid-like shape with membrane blebbing, and thus preventing increased cell migration. Intercepting EV-mediated intercellular communication in the tumor niche with an anti-CD9 Fab antibody, combined with direct targeting of cancer cells, could lead to the development of new anti-cancer therapeutic strategies.

## 1. Introduction

Intercellular communication between cancer and healthy cells is now recognized as an important aspect of tissue transformation that would promote the growth of cancer and the dissemination and seeding of cancer cells in pre-metastatic sites. Nanosized extracellular vesicles (EVs) appear to play a major role in these processes [1]. They carry biological information (e.g., membrane and soluble proteins, lipids, metabolites, and nucleic acids—notably messenger RNA, microRNA, and long non-coding RNA) that reflect, at least in part, the characteristics of donor cells [2,3]. These bioactive molecules could act as mediators in the regulation of neighboring and distant host/recipient cells. EVs are released either by a membrane budding process occurring at the plasma membrane or by fusion of late endosomal multivesicular bodies (LE/MVBs) with the plasma membrane, resulting in the release of their intralumenal vesicles [4,5,6]. EVs are then referred to as microvesicles/ectosomes and exosomes according to the first and second mode of release, respectively. Once discharged into the extracellular medium, such as a biological fluid, in addition to their potential degradation, EVs may be trapped in the extracellular matrix and/or taken up by surrounding cells [7]. The direct interaction between EVs and cells has become the basis of a short- or long-distance intercellular communication mechanism where EVs can trigger a cellular response in host cells. Thus, EVs can reprogram their fate, by promoting their proliferation or differentiation, as well as stimulate their migration among various cellular processes [8,9,10].

The fusion of EVs and cell membranes and/or various mechanisms of endocytosis are described to explain the transfer of EV cargoes [11,12,13,14]. Direct binding of EVs to cells, similar to ligand–receptor interaction, could also occur and promote cell signaling. EVs as mediators of intercellular communication mechanisms are of general interest in various fields, as the molecular transfer of active biomolecules is involved in development, enables homeostasis, and is often dysregulated in various diseases, including cancers [15]. In the latter case, the increased release of cancer cell-derived EVs may not only promote cancer growth but also metastasis [16,17,18] (reviewed in Ref. [19]). In such a context, the interception of EV-mediated communication could find an application in oncology.

Inhibition of the biogenesis and release of EVs from donor cells could be a possibility to counteract their effect in cancer. For instance, proteins involved in exosome formation and/or secretion, such as neutral sphingomyelinase 2 and Rab27a, could be targeted, resulting in a reduction of certain exosome subpopulations [20,21,22,23]. Similarly, the intracellular calcium (Ca^2+^) pool may also be altered, resulting in a defect in EV release from cancer cells [24,25]. Other regulators and potent drugs of exosome secretion have also been identified [26] (reviewed in Ref. [27]). Alternatively, it is possible to impede EV uptake into receptor/target cells by interfering with certain modes of endocytosis, such as clathrin-dependent or independent mechanisms, e.g., the lipid raft/caveolin-mediated pathways using chemical drugs. The uptake of EVs by phagocytosis or micropinocytosis can also be intercepted when these mechanisms occur. For a list of inhibitors and their molecular targets/pathways, readers are referred to the following review [13]. However, the general use of these inhibitors is limited because several mechanisms of EV internalization may be involved concurrently, and systemic application of these drugs, which often act on various cellular pathways, could result in toxic side effects.

Other therapeutic strategies to remove or neutralize cancer cell-derived EVs from circulation have been suggested (reviewed in Ref. [28]). For example, Nishida-Aoki and colleagues proposed to capture circulating EVs derived from cancer cells using specific antibodies (Abs) against EV-associated proteins [29]. Treatment of mice with anti-CD9 (see below) or anti-CD63 Abs stimulated EV clearance by macrophages. Although this treatment had no effect on the primary tumor, tumor metastasis was significantly reduced. Thus, elimination of cancer-derived EVs may be a novel therapy strategy for cancer metastasis. Targeting EV surface proteins may also impede their distribution to distant anatomical sites [29,30], as evidenced by the correlation between expression of specific integrins on the surface of EVs and metastatic tropism [7].

In such a context, one of the EV-associated proteins has attracted attention in the literature, namely CD9 (also known as tetraspanin-29, motility-related protein). CD9 is a member of the tetraspanin superfamily that, by interacting with various protein partners, has various cellular functions, such as cell–cell contact, cell–extracellular matrix interaction, integrin-dependent cell migration, and membrane fusion, among others (reviewed in Refs [31,32,33]). In addition to localizing at the cell surface, where it orchestrates membrane organization, CD9 is associated with EVs including exosomes [34,35]. Of note, a nuclear pool of CD9 was also reported [36]. It has been shown that the presence of CD9 on the surface of EVs and/or on the plasma membrane of recipient breast cancer cells is essential for the uptake of EVs, as its silencing by RNA interference prevents internalization [37]. CD9 is also important for sperm–egg fusion as CD9-deficient oocytes do not fuse properly with sperm during fertilization, thus reduced female fertility is consequently observed in CD9 knockout mice [38,39]. Interestingly, sperm fusion properties are conferred by the CD9^+^ EVs released from eggs [40], and Abs directed against the extracellular domains of CD9 inhibited sperm–egg interaction and fusion [41]. Anti-CD9 Abs can also block the transfer of molecules between CD9^+^ EVs in the epididymal fluid and maturing spermatozoa [42]. Caution should be applied when divalent anti-CD9 Abs are proposed for clinical application. Being the major platelet surface protein [43], CD9, together with the fibrinogen receptor α2bβ3, can trigger platelet activation, aggregation, or lysis, depending on the Ab subclass used [44]. Therefore, the clinical development of such biological tools should exclude the occurrence of potential toxic events, including severe thrombocytopenia and/or thrombocyte aggregation [45,46,47,48] (reviewed in Ref. [33]).

Recently, we designed a CD9-based strategy to block EV transfer between cells using fragment antigen-binding fragments (Fab fragments; hereafter Fab) generated from 5H9 Ab (CD9 Ab) directed to human CD9 [49,50]. We showed that monovalent CD9 Fab impedes the internalization of melanoma cell-derived EVs and nuclear transfer of their cargo proteins in recipient cells [50]. Here, we investigated whether this approach could have therapeutic utility for intercepting the EV-mediated transformation of colon carcinoma. To this aim, we used the established isogenic cell line model of non-metastatic SW480 cells and highly metastatic SW620 cells, where EVs derived from the latter can transform the former and impact their malignant properties, including their motility [10,51]. These cell lines were derived from primary (i.e., SW480 cells) and secondary (SW620 cells) tumors from a single patient [52,53,54]. Our data reveal that monovalent CD9 Fab, but not divalent CD9 Ab, blocks the EV-mediated increase in malignancy of colorectal cancer cells.

## 2. Materials and Methods

### 2.1. Cell Culture

Human SW480 (CCL-228^TM^) and SW620 cells (CCL-227^TM^) were obtained from the American Type Culture Collection (ATCC, Manassas, VI, USA). They were cultured in RPMI-1640 medium (catalog number (#) 10-041-CV, Corning Inc., Corning, NY, USA) supplemented with 10% fetal bovine serum (FBS, #26140079), 2 mM L-glutamine (#25030081), 100 U/mL penicillin, and 100 μg/mL streptomycin (#15140122), all from Thermo Fisher Scientific (Waltham, MA, USA), and incubated at 37 °C in a 5% CO_2_ humidified incubator. Both cell lines were regularly tested for mycoplasma contamination by polymerase chain reaction using the MycoSEQ™ Mycoplasma Detection Kit (#4460626, Thermo Fisher Scientific) according to the manufacturer’s protocol, or upon staining with 4’,6-diamidino-2-phenylindole (DAPI; #D9542, Sigma-Aldrich, St. Louis, MO, USA) and visualization under an Eclipse TE2000-U inverted fluorescence microscope (Nikon, Melville, NY, USA).

### 2.2. Lentiviral Infection

To inhibit the expression of human CD9 (NCBI protein accession number: P21926), transduction-ready CD9 short hairpin (sh) RNA lentiviral particles (#sc-35032-V, Santa Cruz Biotechnology, Dallas, TX, USA) were employed. Viruses were loaded on non-tissue culture treated 24-well plates (#15705-060, VWR International, Radnor, PA, USA) coated with 50 µg/mL of RetroNectin^®^ recombinant human fibronectin fragment (#T100B, Takara Bio USA, San Jose, CA, USA), then centrifuged at 960× *g* for 30 min at 4 °C. The supernatant was removed and plates were washed with PBS before addition of SW480 cells. CD9-deficient (shCD9) SW480 cells were then selected by introducing 2 µg/mL of puromycin into the culture medium for a week. The antibiotic was removed three days before the start of the experiments.

### 2.3. Plasmid and Transfection

SW620 cells were transfected with pCMV6-AC-GFP plasmid encoding for CD9 with a GFP tag at its C-terminus under the control of the cytomegalovirus promoter (#RG202000; OriGene Technologies, Rockville, MD, USA) using Lipofectamine 3000 (#L3000008, Thermo Fisher Scientific) in a 1:2 DNA/lipid ratio. Cells expressing the neomycin resistance gene were selected by introducing 400 µg/mL of G418 Sulfate (Geneticin^TM^ Selective Antibiotic, #10131035, Thermo Fisher Scientific) into the culture medium. After selection, at least 98% of the cells expressed CD9-GFP fusion protein, assessed by fluorescence microscopy. The antibiotic was removed three days before the start of the experiments.

### 2.4. Isolation and Characterization of EVs

EVs released from SW480, SW620, or CD9-GFP^+^ SW620 cells (250,000 cells) cultured in a serum-free medium supplemented with 2% B-27 supplement (#17504044, Thermo Fisher Scientific) on a 6-well plate coated with 20 µg/mL of poly(2-hydroxyethyl methacrylate) (#P3932, Sigma-Aldrich) were enriched by differential centrifugation from the conditioned media after 72 h of incubation. Briefly, after low-speed centrifugations (300 and 1200× *g*) of the conditioned medium, the supernatant was centrifuged at 10,000× *g* for 30 min. The resulting supernatant was centrifuged at 200,000× *g* for 60 min. All centrifugation steps were performed at 4 °C. The 200,000 *g*-pellet was resuspended in 200 µL PBS and stored at −80 °C in small aliquots. The concentration and size of EVs were evaluated by nanoparticle tracking analysis (NTA). We used the light-scattering characteristics of 488 nm laser light on EV preparations injected into the sample chamber of the ZetaView unit (software v8.05.10, Particle Metrix GmbH, Meerbusch, Germany). The calculated EV concentration and size were an average of 11 positions across the analysis window, each recorded in 2 s videos (30 frames per second). Camera gain and minimum trace length were set to 10 and 15, respectively. The concentrations for SW480, SW620, and CD9-GFP^+^ SW620 EVs were 7.3 × 10^10^, 6.9 × 10^10^, and 7.6 × 10^10^ particles/mL, respectively.

EVs were characterized either by immunoblotting for the presence of tetraspanin proteins CD9, CD63, CD81, and ALG-2-interacting protein X (ALIX) or the absence of endoplasmic reticulum (ER)-associated calnexin according to the guidelines of the International Society of Extracellular Vesicles (MISEV2018) [55] or by high-resolution direct stochastic optical reconstruction microscopy (dSTORM) (see below). All relevant data concerning our EV characterization were submitted to the EV-TRACT knowledgebase (EV-TRACK, https://evtrack.org, ID: EV220040, accessed on 13 January 2022) [56].

### 2.5. CD9 Antibody Fab Fragment

The culture of 5H9 hybridoma cells [49] and the production of mouse monoclonal CD9 Ab were recently described [50]. The corresponding CD9 Fab was generated using the Pierce Fab Purification kit (#44985, Thermo Fisher Scientific). Herein, CD9 Ab (500 µg) was incubated with papain immobilized on agarose resin for 3 h at 37 °C. The digested Abs were collected by centrifugation (5000× *g*, 1 min) using a spin column, and the flow through containing the Abs was placed in a new tube. The fragment crystalline (Fc) fragment was removed from digested Ab samples by centrifugation (1000× *g*, 10 min) using the NAb Protein A Plus Spin Column, and the flow through containing the purified Fab fraction was collected. The column was then washed twice with PBS and each wash fraction was combined with the Fab fraction. Using the Microsep Advance Centrifugal Device (10K molecular weight cut-off) purchased from Pall Corporation (#MCP010C46, Westborough, MA, USA), the Fab fraction was concentrated by spinning at 3000× *g* for 25 min at 4 °C. Concentration was then measured by absorbance at 280 nm (final yield of 0.4–0.8 mg/mL). CD9 Fab preparation was assessed by sodium dodecyl sulfate-polyacrylamide gel electrophoresis (SDS-PAGE) and Coomassie blue staining as described in [50].

### 2.6. Immunoblotting

Cells were lysed in cold buffer containing 1% Triton X-100 (#X100, Sigma-Aldrich), 100 mM NaCl, and 50 mM Tris-HCl with a pH of 7.5 and supplemented with Protease Inhibitor Cocktail Set III (#539134, Sigma-Aldrich), followed by incubation on ice for 30 min. Samples were then centrifuged at 12,000× *g* for 10 min at 4 °C. Detergent lysate was collected and Laemmli sample buffer (#1610747, Bio-Rad, Hercules, CA, USA) containing β-mercaptoethanol (#444203, Sigma-Aldrich) was added. The reducing agent was omitted for CD9, CD63, and CD81 immunoblots. For the analysis of EVs, the Laemmli buffer was added directly to the enriched EVs. All samples were heated at 95 °C for 5 min. Proteins were separated by SDS-PAGE using a 4–20% Mini-PROTEAN TGX precast gel (#4561096, Bio-Rad) along with the Trident pre-stained protein molecular weight ladder (#GTX50875, GeneTex, Irvine, CA, USA) and were then transferred to a nitrocellulose membrane (#88018, Thermo Fisher Scientific) overnight at 4 °C. Membranes were incubated in the blocking buffer (PBS containing 1% bovine serum albumin (BSA, #001-000-161, Jackson ImmunoResearch Laboratories Inc., West Grove, PA, USA)) for 60 min at room temperature (RT) and then probed with primary Abs (see below) for 60 min at RT. After 3 washing steps of 10 min each with PBS containing 0.1% Tween 20, membranes were incubated with fluorescein (FITC)-conjugated donkey anti-mouse IgG (1:100, #715-095-150, Jackson ImmunoResearch Laboratories) for 30 min at RT. Finally, membranes were washed thrice (10 min each) in PBS containing 0.1% Tween 20, rinsed in deionized H_2_O, and antigen-Ab complexes were visualized in the iBright FL1000 system (Thermo Fisher Scientific).

Blots were probed with mouse monoclonal anti-CD9 (clone 5H9, 1:500, see above) [49] or anti-CD63 (clone Ts63, 1:500), anti-CD81 (clone 1.3.3.22, 1:500), and anti-Calnexin Abs (clone AF18, 1:500), all purchased from Thermo Fisher Scientific (#10628D, MA5-13548, and MA3-027, respectively), or with anti-Alix Ab (clone 3A9, 1:500, #2171, Cell Signaling Technology, Danvers, MA, USA) and anti-β-actin Ab (clone C-2, 1:1000, #sc-8432, Santa Cruz Biotechnology).

### 2.7. Cell–EV Incubation

SW480 cells or shCD9 SW480 cells (1 × 10^5^) seeded in 1 mL cell medium on poly-D-lysine-coated 35 mm dishes containing #1.5 glass coverslips (#P35GC-1.5-14-C, MatTek Corporation, Ashland, MA, USA) were pre-incubated with various concentrations (6.25, 12.5, and 25 µg/mL) of CD9 Fab or divalent CD9 Ab for 30 min at 37 °C. Concurrently, EVs (1 × 10^9^ particles, equivalent volume of ≈15 µL) derived from SW620 or CD9-GFP^+^ SW620 cells were pre-incubated with the same concentrations of Abs for 30 min at 4 °C. Afterward, EVs were added to the cells and co-incubated for 5 (or 16) h in the presence of Abs. The final EV concentration is 1 × 10^9^ particles/mL or 27 µg protein/mL. These conditions will be referred as protocol #1 (*Cells and EVs*). As control, Abs were omitted. In some experiments, EVs (1 × 10^9^ particles) alone were pre-incubated with CD9 Fab or divalent CD9 Ab as above and then added to cells (protocol #2, *EVs*). Conversely, cells were pre-incubated with CD9 Fab or divalent CD9 Ab as above and EVs (1 × 10^9^ particles), which were not incubated with Abs, were then added and both cells and EVs were incubated together for 5 h (protocol #3, *Cells*). Note that in all conditions, Abs were not removed prior to co-incubation of cells and EVs, resulting in different Ab concentrations during the 5 h co-incubation, especially for protocol #1 (or #3) compared to protocol #2. Alternatively, the Abs were removed from cells after the 30 min pre-incubation and before the addition of EVs which were not pre-incubated with Abs (protocol #3′, *Cells*). These distinct protocols are summarized in Appendix A. In other experiments, cells were pre-treated with or without 10 µM PRR851, a drug synthesized in one of our laboratories (for detail, see Ref. [57]), in the presence or absence of CD9 Fab (25 µg/mL) for 30 min at 37 °C. CD9-GFP^+^ EVs were then added for 5 h. DMSO alone was used as vehicle control. Cells were then fixed and prepared for immunocytochemistry.

### 2.8. Confocal Laser Scanning Microscopy

Cells grown on poly-D-lysine-coated dishes, as described above, were fixed in 4% paraformaldehyde (PFA) in PBS for 15 min, washed twice with PBS, and permeabilized with 0.2% Tween 20 in PBS (permeabilization buffer, PB) for 15 min. They were first blocked with 1% BSA diluted in PB for 30 min and then immunolabeled, using either rabbit antiserum directed against the SUN domain-containing protein 2 (SUN2) (1:50, #PA5-51539, Thermo Fisher Scientific), which labels the inner nuclear membrane (INM), or mouse monoclonal CD9 Ab (clone 5H9, see above), to label the cell membrane for 60 min. In some experiments, cells were not permeabilized. All steps were performed at RT. Afterward, cells were washed twice with PBS and incubated with Alexa Fluor^®^647-conjugated goat anti-rabbit IgG (1:1000, #A-21246) or Alexa Fluor^®^488-conjugated goat anti-mouse IgG (1:1000, #A11017), both from Thermo Fisher Scientific, for 30 min. The washing step was repeated prior to observation. Primary and secondary Abs were diluted either in PB containing 1% BSA or, in the case of non-permeabilized immunolabeling, in PBS with 1% BSA. To assess membrane rounding and blebbing, PFA-fixed cells were instead stained with Alexa Fluor488^®^-conjugated Phalloidin (#A12379, Thermo Fisher Scientific) for 40 min to label actin. Nuclei were counterstained with DAPI. Images were acquired by confocal laser scanning microscopy (CLSM) using the Nanoimager S Mark II system (Oxford Nanoimaging (ONI), Oxford, UK) with 100× oil-immersion objective under constant microscope settings. A total of 20 x-y optical sections of 0.45 µm thickness were acquired for each cell of interest. Raw images were processed using Fiji. To measure GFP fluorescence derived from endocytosed CD9-GFP^+^ SW620 cell-derived EVs in the cytoplasm, regions of interest (ROIs), excluding the DAPI-stained nucleus, were drawn around the plasma membrane using corresponding bright-field images (not shown) as a guide. Total cell fluorescence was then determined using the “measure” function in Fiji across all optical sections. To count nuclear fluorescent materials, each optical section through the cell was assessed individually by drawing ROIs around the SUN2-labeled nuclear membrane. An auto threshold was then applied and, using the “analyze particle” function in Fiji, signals showing greater than 8 pixel counts were considered as positive and the results from all sections were combined.

### 2.9. Stochastic Optical Reconstruction Microscopy

Direct stochastic optical reconstruction microscopy (dSTORM) was applied on EVs derived from SW480, SW620, and CD9-GFP^+^ SW620 cells. EVs were immunolabeled and imaged using the EasyVisi Single-Extracellular Vesicle Characterization kit from ONI (beta v1.0, Oxford Nanoimaging, UK) as described previously [57]. Briefly, EVs (3.5 × 10^7^ particles) were incubated overnight at 4 °C with the fluorescently labeled Abs: CD9-Atto488, CD63-Cy3B, and CD81-AlexaFluor^®^647. Labeled EVs were then immobilized on microfluidic chips coated with PEG-Biotin. All preparations were performed on a Roboflow automated system platform (ONI). Freshly prepared BCubed STORM-imaging buffer was added prior to image acquisition. Labeled proteins were imaged sequentially at 45%, 50%, and 50% power for the 647, 561, and 488 nm lasers, respectively, at 2000 frames per channel with the angle of illumination set to 52.5°. Prior to the start of the imaging session, channel mapping was calibrated using 0.1 µm TetraSpeck beads (#T7279, Thermo Fisher Scientific). Data were processed on NimOS software (v1.18, ONI, Oxford, UK). Note that under these conditions, GFP fluorescence is not detected (data not shown). To identify subpopulations of EVs that express one, two, or three markers, images were analyzed using ONI’s online platform called CODI (https://alto.codi.bio/, release versions 0.20 to 0.24; July to October 2021, accessed on 21 September 2021). Density-based clustering analysis with drift correction was then performed to evaluate each vesicle.

### 2.10. Flow Cytometry

To determine the number of cell surface CD9 molecules, the Quantum^TM^ Simply Cellular^®^ (QSC) anti-mouse IgG kit (#815, Bangs Laboratories Inc., Fishers, IN, USA) was utilized. SW480 and SW620 cells (1 × 10^5^) and 4 microsphere populations containing increasing levels of Fc-specific capture Ab were incubated with FITC-conjugated anti-CD9 Ab (clone eBioSN4, 1:20, #11-0098-42, Thermo Fisher Scientific) in PBS containing 0.5% BSA for 30 min on ice. Both cells and microspheres were then analyzed by the CytoFlex flow cytometer (Beckman Coulter, Indianapolis, IN, USA) using the same detector gain for all samples. A standard curve was generated using the median channel values of the microspheres, and the amount of CD9 molecules per cell was determined from this curve. All calculations were performed using the QuickCal^®^ program (v2.3, www.bangslabs.com, accessed on 13 October 2021).

### 2.11. Cell Migration

*Scratch wound healing assay*—Cell migration was evaluated by a scratch wound healing assay. Briefly, SW480 cells were seeded at a concentration of 2 × 10^5^ cells/well in 12-well standard cell culture plates (#83.3921, Sarstedt Inc., Nümbrecht, Germany) and, after reaching 100% confluence, a scratch was introduced on the cell monolayer with a sterile pipet tip. The detached cells were washed with PBS. Cells and/or SW620 cell-derived EVs were then pre-incubated for 30 min with CD9 Fab or divalent CD9 Ab at various concentrations (6.25, 12.5, and 25 µg/mL) according to protocols #1 to #3. Afterward, cells and EVs were co-incubated for 5 h. In all conditions, Abs were not removed during cell–EV incubation. As positive control, cells were exposed to EVs without incubation with Ab. Images of scratch wounds were captured using an inverted Olympus IX70 microscope (Olympus Italia S.r.l, Segrate, Italy) before (0 h) and after (5 h) the addition of EVs. Wound areas were measured by ImageJ software [58]. The wound area at 0 h was considered as baseline (100%). The wound healing assays were performed at least five times for each condition.

*Transwell filter assay*—SW480 or SW620 cells were grown to 80% confluency then serum starved for 24 h in DMEM-F12 cell medium (#12634010, Thermo Fisher Scientific) supplemented with 0.5% FBS at 37 °C. The migration assay was performed using 8 µm pore size 24-well Transwell plates (#3464, Corning Inc.). Briefly, the lower chamber was filled with 800 µL of cell medium as described above, followed by the addition of starved 1 × 10^5^ cells in 200 µL medium to the upper chamber. For SW480 cells, they were allowed to attach for 3 h prior to incubation with 25 µg/mL CD9 Fab or divalent CD9 Ab for 30 min at 37 °C. Concurrently, SW620-derived EVs (1 × 10^9^ particles) were incubated with the same concentration of Abs for 30 min at 4 °C according to protocol #1, then co-incubated with the cells for 24 h at 37 °C. As controls, either EVs or Abs were omitted or cell medium contained 10% FBS (data not shown). In experiments involving SW620 cells, the same procedure was performed as above without the addition of EVs. After the 24 h incubation, the number of migrated cells was evaluated as previously described with minor modifications [59]. Briefly, cells on the upper chamber were carefully removed using cotton swabs, and the migrated cells adhered to the bottom side of the microporous membrane, as well as those in the lower chamber, were detached by trypsinization. Cells were collected, pelleted by centrifugation, and resuspended in the residual medium. The number of invasive cells was determined by automated counting using the TC20 automated cell counter (Bio-Rad).

### 2.12. Statistical Analysis

All experiments were performed at least in triplicate. Data are presented as the mean ± standard deviation (S.D.). Statistical analysis was determined using a two-tailed Student’s *t*-test, and *p* values < 0.05 were considered significant. All graphs were created using GraphPad Prism 8 (v8.4.3, Dotmatics, Boston, MA, USA).

## 3. Results

To investigate whether a monovalent Fab generated from mouse monoclonal CD9 Ab (clone 5H9) directed to human CD9 [49,50] impedes EV-mediated morphological transformation of colon cancer cells, we used the established model of non-metastatic SW480 and highly metastatic SW620 cells [52,53]. One of our laboratories has previously shown that morphological traits of SW620 cells can be transferred to SW480 cells via EVs [10,52,53]. Labeling of these cells with CD9 Ab confirmed their distinct morphologies; SW480 cells are flat with a spread mesenchymal-like shape and tend to form multicellular clusters, whereas SW620 cells are rounded and have numerous membrane blebs reminiscent of an amoeboid phenotype (Figure 1A). In both cell lines, CD9 antigen is present on the cell surface and in the cytoplasmic compartment (Figure 1A). The number of cell surface CD9 molecules per cell was quantified by flow cytometry after labeling with fluorochrome-conjugated anti-CD9 Ab (clone eBioSN4, see Section 2), while the total amount of CD9 was quantified by immunoblotting (Figure 1B,C). Both approaches revealed that CD9 protein is significantly more expressed in SW480 compared to SW620 cells. These observations are in agreement with the previously reported CD9 transcript level for these cells [60]. These data suggest, albeit indirectly, that CD9 per se is not responsible for the morphological difference between SW480 and SW620 cells, although some level of its expression may regulate the function of its interacting partners [10,52,53].

To monitor EV-mediated cell transformation, we engineered SW620 cells to express the CD9-GFP protein that would produce fluorescent EVs (Figure 1D,E). As observed by GFP fluorescence, CD9 overexpression did not alter the morphology of SW620 cells, with their rounded appearance and membrane blebs remaining present (Figure 1D).

### 3.1. Characterization of EVs Released by SW620 Cells

Next, for comparison, we characterized the EVs released by SW620 cells as well as those produced by CD9-GFP^+^ SW620 and SW480 cells. The EVs were enriched from the conditioned media by differential ultracentrifugation (for technical detail, see Section 2). First, we measured their size by NTA. The EVs derived from the SW620, CD9-GFP^+^ SW620, and SW480 cell lines showed a similar diameter of 148 ± 1.9, 157 ± 9.4, and 144 ± 2.5 nm (mean ± S.D., *n* = 3 independent measurements), respectively (Figure 2A, pink area). Interestingly, large EVs with a size of 350–500 nm were preferentially observed in those produced by SW480 cells (Figure 2A, grey area). Second, the expression of bona fide EV markers was determined by immunoblotting in EVs released by SW620 cells. The tetraspanin membrane proteins CD9, CD63, CD81, and cytoplasmic ALIX were detected therein (Figure 2B). Alix was demonstrated to be involved in the sorting of tetraspanins to exosomes [61]. As expected, the ER-associated calnexin was absent (Figure 2B). These data are in agreement with our previous studies, as well as studies by others [57,62,63,64]. The overexpression of CD9-GFP did not influence the presence or absence of these proteins in SW620 cell-derived EVs (Figure 2C). Third, the distribution of tetraspanins among the EVs was determined at super-resolution using dSTORM (Figure 2D). Analysis of small EVs (<200 nm in diameter) revealed the heterogeneity of the EV population with a variable distribution of single, double, and triple positives (Figure 2E and Appendix A), in agreement with previous studies [57,65]. Triple-positive EVs (CD9, CD63, and CD81) constituted the major population (>70%) of EVs. They could correspond to exosomes arising from LE/MVB fusion with the plasma membrane [34]. Single-positive EVs represented minor or negligible fractions among the EV subpopulation, irrespective of the CD marker. Of note, overexpression of CD9-GFP in SW620 cells slightly reduced the amount of CD63^+^CD81^+^ EVs with a concomitant increase in triple-positives. In SW620 cells, CD63^+^CD81^+^ EVs constituted about 17 ± 10% of total EVs. These EVs cannot be targeted by anti-CD9 Ab. In contrast, single CD9^+^ EVs are highly enriched in large EVs (350–500 nm in diameter) and represent the major fraction (>80%) (Appendix A). They could correspond to microvesicles/ectosomes derived directly from the plasma membrane. Double- or single-positive CD63 and CD81 were not detected. Thus, all large EVs contain CD9. Overall, the complexity of EV subtypes (exosomes versus microvesicles) and their subpopulations, as monitored at an individual level, indicates that CD9^+^ EVs account for the majority of EVs.

### 3.2. Internalization of SW620 Cell-Derived CD9-GFP^+^ EVs into SW480 Cells: Impact of CD9 Ab

By applying fluorescent EVs released by CD9-GFP^+^ SW620 cells to recipient SW480 cells for a period of 5 h, we observed, after cell fixation and DAPI staining, the internalization of CD9-GFP, as detected by green fluorescence in the cytoplasmic compartment (Figure 3A, control). To assess the impact of anti-CD9 Abs on EV internalization, we pre-incubated both cells and EVs with either monovalent CD9 Fab or divalent Ab (25 µg/mL) for 30 min and then co-incubated them for 5 h in the presence of Abs. This is the first of three protocols (#1-3) used in this study, which are summarized in Appendix A (see below and Section 2). Interestingly, application of CD9 Fab, but not CD9 Ab, reduced the amount of GFP in the cytoplasmic compartment as observed by CLSM (Figure 3A, CD9 Fab). Through a series of x-y optical sections covering the entire cell of interest, quantification revealed a significant decrease in cytoplasmic GFP fluorescence in CD9 Fab-treated cells, whereas an increase occurred in those incubated with CD9 Ab (Figure 3B). The effect of CD9 Fab depends on the concentration used (Figure 3B).

We recently demonstrated that EV-associated CD9 can reach the nucleus after EV internalization and their endocytic transport, where EV-containing late endosomes enter the type II nuclear envelope invaginations of the nucleoplasmic reticulum [37] (reviewed in Ref. [14]). The latter process involved a protein complex (named VOR) formed by vesicle-associated membrane protein (VAMP)-associated protein A (VAP-A), which is localized at the outer nuclear membrane, oxysterol-binding protein (OSBP)-related protein-3 (ORP3), and endosomal Rab7 [66]. Their interactions can be inhibited by a novel chemical drug, PRR851, which blocks Rab7 binding to VAP-A-ORP3 and thereby prevents nuclear transfer of the EV-associated protein [57]. To visualize the nuclear transfer of the EV cargo and the impact of CD9 Abs on it, upon 5 h incubation with CD9-GFP^+^ EVs, SW480 cells were fixed, immunolabelled for SUN2 to highlight the inner nuclear membrane, and analyzed by CLSM. In each x-y optical section, the nuclear CD9-GFP appeared as discrete punctate signals (Figure 3C), in which we set a threshold level above eight pixels to exclude (auto)fluorescent signals (see Section 2, Appendix A). Analysis of the 20 sections covering the entire nuclear compartment revealed a reduction in nuclear CD9-GFP in cells and EVs treated with CD9 Fab (Figure 3C and Appendix A). This effect did not seem to occur with those incubated with CD9 Ab. Quantification of these cells confirmed these observations (Figure 3D). Although significant, the CD9 Fab-mediated reduction in nuclear transfer of CD9-GFP appeared moderate compared to EV internalization (compare Figure 3B versus Figure 3D, CD9 Fab). Indeed, fold change analysis indicates that CD9 Fab had more impact on the initial internalization of CD9-GFP than on its nuclear transfer (Appendix A). In contrast, the addition of PRR851, alone or in combination with CD9 Fab, abolished the nuclear transfer of CD9-GFP (Appendix A). This observation suggests that intracellular transport of EV-associated proteins from the plasma membrane to the nucleoplasm of recipient cells via the endocytic compartment is highly efficient (see Section 4).

### 3.3. Pro-Metastatic Morphological Alterations of SW480 Cells by SW620 Cell-Derived EVs Are Blocked by CD9 Fab

To assess whether CD9 Fab negatively interfered with the pro-metastatic morphological alterations of SW480 cells induced by SW620 cell-derived EVs, we co-incubated them in the presence or absence of Abs using three distinct protocols (Appendix A) and then determined their phenotypic modification, i.e., cell rounding and induction of membrane blebs. In the first set of experiments, following protocol #1, both cells and EVs were pre-incubated with either monovalent CD9 Fab or divalent Ab (25 µg/mL) for 30 min. Cells and EVs were then co-incubated for a period of 5 h. SW480 cell morphology was determined by fixing and staining them with fluorochrome-conjugated phalloidin and DAPI to label actin filaments [67] and nuclei, respectively, prior to CLSM analysis. As negative and positive controls, we used SW480 cells and those incubated with SW620 cell-derived EVs, respectively. In the latter, we observed cellular transformation with the appearance of rounded cells and membrane blebbing (Figure 4A). Quantification revealed that both phenotypes were infrequently detected in those not incubated with EVs (Figure 4B,C, respectively). The addition of CD9 Fab (25 µg/mL) significantly reduced EV-mediated cell transformation, a phenomenon also observed at lower concentrations (Figure 4A–C). In contrast, the divalent CD9 Ab did not interfere with such processes (Figure 4A–C). Similar results were obtained when co-incubation of cells and EVs was increased to 16 h (Appendix A).

In the second set of experiments, following protocol #2 (Appendix A), only EVs were pre-incubated with monovalent CD9 Fab or divalent Ab for 30 min, then added to cells and co-incubated for 5 h. Again, cell rounding and membrane detachment were both inhibited by the addition of CD9 Fab but not CD9 Ab (Appendix A), suggesting that saturation of EV-associated CD9 interferes with these EV-mediated processes. Finally, similar data were obtained when only cells were pre-incubated with either monovalent or divalent CD9 Abs, according to protocol #3 (Appendix A), prior to their 5 h co-incubation (Appendix A). Note the presence of CD9 Abs during cell–EV incubation, especially in protocol #3, where free Abs would bind to EVs and could therefore have an impact on the result. To evaluate this, we performed a variant of protocol #3 (#3′) where unbound CD9 Abs were removed from the cells after the 30 min pre-incubation and before the addition of EVs (Appendix A). Interestingly, under these conditions, the pre-metastasis morphological alterations of SW480 cells were not significantly impacted by the addition of CD9 Fab (Appendix A), suggesting that the saturation of CD9 present in EVs is essential to interfere with their action.

### 3.4. SW620 Cell-Derived EV-Induced Migration of SW480 Cells Is Impeded by CD9 Fab

SW620 cells have a more rounded shape and strong plasma membrane blebbing activity (see above Figure 1A,D), which is associated with amoeboid motility [68]. Since these morphological traits can be transferred to SW480 cells via EVs derived from SW620 cells and influence their motility [8,9,10,51], we evaluated their migration and the impact of CD9 Abs. For this purpose, we used two classical methods: the linear wound healing assay and the Transwell filter assay (see Section 2). First, the wound healing assay, performed for 5 h, revealed that SW620 cell-derived EVs stimulated the motility of SW480 cells (Figure 5A), which is in agreement with our previous study [57]. Interestingly, exposure of cells and EVs or only EVs to CD9 Fab according to protocol #1 or #2, respectively, before their co-incubation for 5 h had a highly negative effect on their migration (Figure 5A,B). In contrast, divalent CD9 Ab either did not prevent this migration or stimulated it, which is consistent with the data regarding morphological alterations (Figure 4 and Appendix A). Pre-incubation of cells with CD9 Fab or CD9 Ab, as described in protocol #3, did not block cell migration (Figure 5C). The latter data contrast with those observed for the impact of CD9 Fab on morphological alterations (protocol #3, Appendix A) but are consistent with those observed when CD9 Abs were removed prior to the addition of EVs (protocol #3′, Appendix A), again emphasizing the importance of intercepting EVs with CD9 Fab. In protocol #3, we could not exclude that a fraction of EVs reaches some recipient cells before being neutralized by CD9 Fab, and thus stimulates the cellular transformation impacting their migrations.

Second, we used a Transwell membrane filter with 8 µm pores allowing migration of cells from the upper to the lower chamber (Figure 5D). After attachment of the cells to the membrane, we added SW620 cell-derived EVs and co-incubated them for 24 h. The trans-membrane migrated cells recovered in the lower chamber revealed the stimulation of motility for SW480 cells primed by EVs (Figure 5E). The experiment was then repeated by pre-incubating the cells and EVs with CD9 Fab or CD9 Ab (25 µg/mL) for 30 min before their co-incubation (Figure 5D). In agreement with EV-mediated morphological transformation, addition of CD9 Fab reduced the number of migrating cells, whereas divalent Ab increased it, although not to a statistically significant extent (Figure 5E).

Finally, caution should be considered with these assays, as horizontal CD9 binding to adhesion and/or integrin molecules, as well as certain lipids, can suppress or promote cell–cell and cell–extracellular matrix interactions (and consequently cell motility), as these processes potentially being dependent on the cellular system [69,70,71,72] (reviewed in Ref. [33]). This prompted us to evaluate the impact of CD9 Abs on SW620 cell migration. As evaluated with the Transwell filter assay, highly metastatic SW620 cells are migrating more than SW480 cells under the same culture conditions (Figure 5F), as recently reported in [73,74]. Indirectly, these data indicated that CD9 expression levels in SW480 and SW620 cells inversely correlate with migration behavior, suggesting a negative impact of CD9 on cell migration in such SW480/620 cell systems. Interestingly, addition of CD9 Fab to SW620 cells prevented their migration, similar to EV-primed SW480 cells, whereas, unlike the latter whose migration was partly stimulated by divalent CD9 Ab, this Ab further inhibited SW620 cell migration (Figure 5F). The differential response to divalent CD9 Ab suggests that EV uptake and subsequent cellular transformation of SW480 cells is the primary cause of their migration. Further research is needed to determine whether a certain CD9 threshold impacts SW480/620 cell migration (and their morphology, see above), which is beyond the scope of this study.

### 3.5. CD9 Is Essential for Pro-Metastatic Morphological Alterations of SW480 Cells Mediated by SW620 Cell-Derived EVs

All data described so far suggest that CD9 Fab blocks EV-mediated transfer of morphological traits from SW620 to SW480 cells. However, we do not know whether CD9 per se is involved in this process, particularly in the initial binding of EVs to the surface of SW480 cells. To address this issue, we silenced its expression in SW480 cells using shRNA technology. As shown by immunoblotting (Figure 6A), approximately 91 ± 2.7% (*n* = 3) of CD9 is reduced in shCD9 cells compared to parental cells. CD9 knockdown did not affect the overall mesenchymal-like spreading morphology of SW480 cells (data not shown), suggesting that CD9 is not the only player involved in this phenotype (see Discussion). We then co-incubated them with SW620 cell-derived EVs for 5 h and determined their morphological alterations. Interestingly, EV-induced rounded morphology and membrane blebbing were prevented in shCD9 cells (Figure 6B,C, respectively), suggesting that cell-associated CD9 is involved in the initial binding of CD9 EVs and/or their internalization and upstream events.

## 4. Discussion

It is now well recognized that the growth of cancer and its metastases at distinct sites involve not only intrinsic factors, but also extrinsic ones in which EVs secreted either by the cancer cells themselves or by surrounding resident cells, such as mesenchymal stromal cells and fibroblasts, play a role [7,35,75]. This bidirectional intercellular crosstalk contributed to the establishment of tumor microenvironmental niches that favor cancer cells over resident cells. Therefore, the development of therapies that target the intercellular communication and metastasis process may find clinical application (reviewed in Refs [76,77]). In this context, we have recently developed a novel approach, based on human CD9 protein, where a monovalent anti-CD9 Fab intercepts the internalization of CD9^+^ EVs by recipient cells such as mesenchymal stromal cells [50].

Here, we report that CD9 Fab can intercept the internalization of CD9^+^ EVs derived from highly metastatic SW620 colon cells to non-metastatic SW480 cells, thereby blocking their cellular transformation. The presence of CD9 at the surface of recipient cells is also important for promoting EV uptake. Among the pro-metastatic properties transferred by SW620 cell-derived EVs are general alterations in cell structure, e.g., a shift from a mesenchymal-like spreading morphology to an amoeboid shape, which promotes conversion of migration from a mesenchymal to amoeboid mode [10,78]. This change is called the mesenchymal–amoeboid transition, which responds to a change in the cellular microenvironment [79]. As shown by the linear wound healing assay and Transwell filter assay, CD9 Fab prevented increased cell motility of EV-primed SW480 cells or impeded SW620 cell migration. In this context, one of our laboratories has previously shown that EVs derived from SW620 cells are enriched in cytoskeleton-associated proteins as well as RhoA interactors that activate the RhoA/ROCK pathway, which is known to induce amoeboid cell migration [10]. This pathway and possibly others, including those that involve the nuclear transfer of EV content (see below) [57], could explain the morphological transformation of SW480 cells. The Akt/mTOR pathway could also be involved [51].

As demonstrated using CD9-GFP, not only the internalization of EV-derived cargo proteins is blocked by CD9 Fab, but also the fraction of them that is transferred into the nucleoplasm of recipient cells. Nuclear transfer of EV cargo could play a role in the reprogramming of host cells, leading to their transformation. For example, nuclear translocation of epidermal growth factor receptor (EGFR) or androgen receptor (AR) associated with EVs released from prostate cancer cells and taken up by indolent receptor cells (i.e., cells without EGFR/AR) activated distinct signaling pathways in the latter [80]. Similarly, we have shown that the transcriptome of mesenchymal stromal cells is altered after their exposure to EVs derived from melanoma cells [37]. Among the genes whose expression is modified are those involved in the inflammation process. More recently, it has been shown that RNA cargoes of EVs released by *Plasmodium falciparum* reach the nucleoli of recipient cells, i.e., monocytes, revealing new aspects of communication, and perhaps function, between pathogen-derived EVs and their host cells [81]. We showed that the nuclear transfer of EV-associated proteins and nucleic acids involved the VOR protein complex [66], which brings together Rab7^+^ late endosomes containing endocytosed EVs and the outer nuclear membrane, with the former entering and/or stimulating the formation of type II nuclear envelope invaginations [37]. Nuclear transfer can be inhibited by a chemical drug, PRR851, which blocks the formation of the VOR complex, i.e., the binding of VAP-A-ORP3 to Rab7, and thus prevents nuclear transfer of the EV-associated protein [57]. Application of PRR851 to SW480 cells before and during their incubation with SW620 cell-derived CD9-GFP^+^ EVs severely impaired the nuclear localization of the fusion protein to a greater extent than that of CD9 Fab alone. Given that PRR851 also blocked the SW620 cell-derived EVs mediated transformation of SW480 cells [57] without impacting the internalization of EVs [57] (this study), the modest reduction in the nuclear transfer of EV cargoes observed with CD9 Fab nevertheless contributed, along with perhaps other factors mentioned above, to the membrane rounding and blebbing phenotypes. This partial inhibition of the nuclear translocation of EV cargoes by Fab CD9 is also interesting despite the significant inhibition of EV internalization at the plasma membrane, suggesting that endocytic transport of the minute fraction of internalized EVs is highly efficient. Loading of late endosomes with EV cargo could facilitate their transport to perinuclear areas and translocation into nuclear envelope invaginations. More investigations are needed to further dissect the mechanisms underlying the transport of EV-loaded endosomes en route to nuclear compartment. By reducing the amount of internalized EVs, CD9 Fab may facilitate such studies.

How does CD9 stimulate cellular internalization of EVs? Although the lateral interactions of CD9 with itself, other tetraspanins, or other membrane proteins forming tetraspanin-enriched microdomains are well described [82], little is known about its potential trans-interaction, i.e., involving CD9 molecules in opposite membranes. However, it has been described that CD9 promotes the clustering of adhesive proteins at the plasma membrane of T cells, antigen-presenting cells, or endothelial cells, e.g., integrin lymphocyte function-associated antigen 1 (LFA-1) and its ligands (intercellular adhesion molecule 1 (ICAM-1) and ICAM-3), and that these regulate cellular interactions at the level of immune synapses [83] or firm adhesion and transendothelial migration of leukocytes [84,85] (reviewed in Ref. [31]). Similarly, CD9 increases cell adhesion mediated by the activated leukocyte cell adhesion molecule (ALCAM, CD166) [86]. In such a context, CD9 Fab could interfere with CD9-mediated clustering of adhesive proteins. It remains to be determined whether CD9–CD9 trans-interaction occurs, as suggested but not proven by divalent CD9 Ab and the impact of CD9 Fab on it. Silencing CD9 in recipient SW480 cells highlighted the implication of this tetraspanin in the cellular internalization of EVs and upstream cellular transformation events. In our previous study, using melanoma or breast cancer cells, we showed that CD9 depletion in EVs also interfered with their cellular uptake [37], indicating that CD9 in both recipient cells and EVs is involved in this process. The lack of major morpho-phenotypic change in SW480 cells depleted of CD9 will deserve additional investigation as upregulation of other tetraspanin proteins might contribute to the overall organization of the plasma membrane, as well as cellular adhesion, allowing the maintenance of their mesenchymal-like morphology in the absence of CD9. In a similar context, an upregulation of CD81 was observed in breast cancer cells upon silencing CD9 [87]. Likewise, the CD9 overexpression, as shown with CD9-GFP in SW620 cells, revealed that other players are essential in establishing a mesenchymal-like morphology. Further studies to examine the impact of CD9 Fab on cell invasion using three-dimensional cultures and/or mouse xenograft models are also needed in the future. This is particularly true for assessing the amount of monovalent Abs and the frequency of injection that would block intercellular communication mediated by cancer cell-derived EVs. In vivo, CD9^+^ EVs released by other cell types, albeit to a lesser extent, can technically neutralize CD9 Fab.

Caution should be taken when targeting CD9 as it may also play a tumor suppressor role in certain cancers and blocking its activity may stimulate cancer progression [88,89,90,91] (reviewed in Ref. [33]). For instance, the examination of surgical tumor samples from patients with colon carcinoma indicated that CD9 was strongly expressed at the primary cells compared to those at metastatic sites in colon carcinoma [60]. This differential expression of CD9 correlated with their invasive properties, in which the divalent CD9 Ab interfered. In our cell system and assays, the situation was more complex when divalent CD9 Ab was used because it either stimulated (although not significantly) or reduced migration as observed for EV-primed SW480 and SW620 cells, respectively, and did not block EV uptake by SW480 cells, but rather enhanced it. This last observation reinforces the principle of not using divalent CD9 Ab for therapeutic purposes (see Section 1). By interacting with various partners, the negative or positive effects of CD9 on cancer cells could depend not only on its own expression level, but also on its binding proteins. In addition, we must consider that CD9 may be involved simultaneously or sequentially in various cellular events (e.g., EV uptake, membrane organization, migration). Thus, interfering with or stimulating its sequestration may favor one function over another. Additional studies are needed to dissect the complete CD9 interactome and determine how the threshold of CD9 (or other tetraspanins and interacting partner proteins) influences cellular transformation and migration processes.

Collectively, we demonstrated that CD9 associated with EVs derived from donor colon cancer cells and present in recipient cells is involved in EV uptake leading to morphological transformation of recipient cells, including the acquisition of aggressive migratory behavior, and that a monovalent Ab directed against this tetraspanin protein can impede its function. Intercepting EV-mediated intercellular communication in the tumor niches (i.e., primary and secondary sites) with an anti-CD9 Fab, combined with direct targeting of cancer cells, could lead to the development of new anti-cancer therapeutic strategies.

## 5. Patents

The United Kingdom patent application GB1814065.7 and United States provisional patents US62724183/US17271690 as well as PTC/EPO applications IB2019/057294/EP19783649.7 are pending. The authors declare no other competing interests.

## Figures and Tables

**Figure 1 cells-11-02474-f001:**
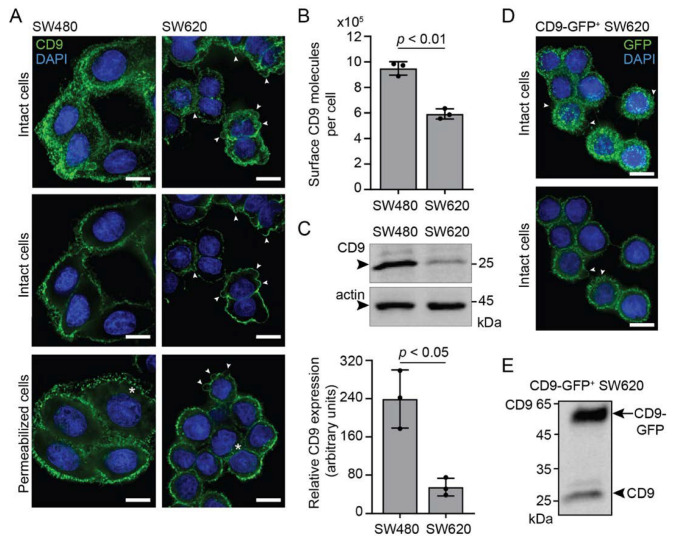
CD9 expression in SW480 and SW620 cells. (**A**) CD9 expression was investigated by indirect immunofluorescence labeling using anti-CD9 5H9 Abs on either intact or permeabilized SW480 and SW620 cells cultured on poly-D-lysine-coated dishes. Nuclei were stained with DAPI and samples were observed by CLSM. Composite images (top panels) or single x-y sections (middle and bottom panels) are shown. Arrowheads indicate membrane blebs, while asterisks mark cytoplasmic CD9 immunoreactivity. (**B**) The amount of surface CD9 antigens per cell detected with FITC-conjugated eBioSN4 Abs was estimated using a flow cytometer calibrated with fluorescent microparticles. (**C**) Total CD9 antigens were analyzed by immunoblotting (top panel) using 5H9 Abs and quantified (bottom panel). The samples were normalized to β-actin. Molecular mass markers (kDa) are indicated. Arrowhead indicates the protein of interest. (**D**,**E**) SW620 cells stably transfected with CD9-GFP were analyzed either by fluorescence microscopy without permeabilization (**D**) or immunoblotting (**E**). For the microscopy, nuclei were stained with DAPI and samples were observed by CLSM. A composite image (top panel) or a single x-y section (bottom panel) is displayed. Arrowheads indicate membrane blebs. For immunoblotting, the membrane was probed with 5H9 Abs. The arrow and arrowhead indicate the CD9-GFP fusion protein and the endogenous CD9, respectively. Means ± S.D. and individual values for each experiment are shown (*n* = 3). *p* values are indicated. Scale bars, 10 µm.

**Figure 2 cells-11-02474-f002:**
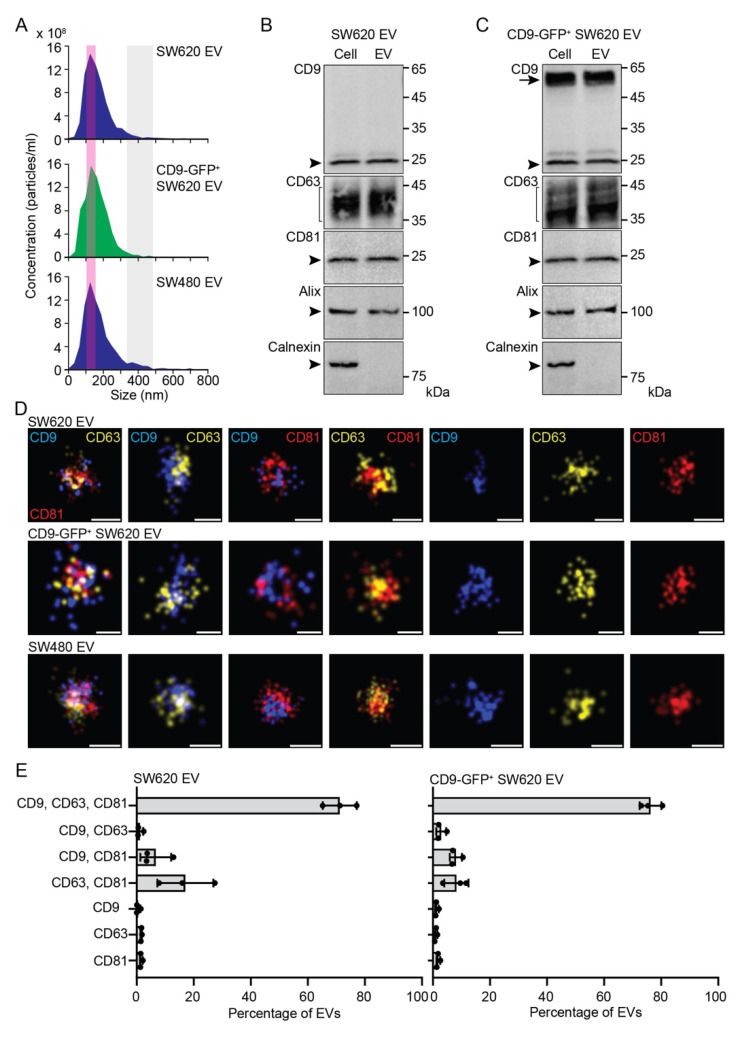
Characterization of EVs released by SW620, CD9-GFP^+^ SW620, and SW480 cells. (**A**–**E**) EVs were recovered from the conditioned media of SW620, CD9-GFP^+^ SW620, and SW480 cells by differential centrifugation, and the resulting 200,000× *g* pellets were analyzed by the ZetaView particle analyzer (**A**), immunoblotting (**B**,**C**), and dSTORM (**D**,**E**). The concentration and size of EVs derived from the indicated cells are shown (**A**). Note the presence of a common population of small particles (<200 nm) with a peak at 100–150 nm (pink area), and larger ones (350–500 nm, gray areas) enriched in SW480 samples. EVs, and for comparison the cells from which they were derived, were probed by immunoblotting for CD9, CD63, CD81, Alix, and Calnexin (**B**,**C**). Arrowheads and brackets indicate the endogenous proteins of interest, while the arrow points to the CD9-GFP fusion protein. Molecular mass markers (kDa) are indicated. EVs were imaged after immunolabeling of three tetraspanins using dSTORM (**D**). The proteins of interest (CD9, CD63, and CD81) were pseudo-colored as indicated. Small single-, double-, and triple-positive EVs were shown (**D**) and quantified (**E**). Means ± S.D. and individual values for the three experiments are shown (*n* > 5000 EVs per experiment). Note that small and large EVs derived from SW480 cells were also quantified (Appendix A). Scale bars, 50 nm.

**Figure 3 cells-11-02474-f003:**
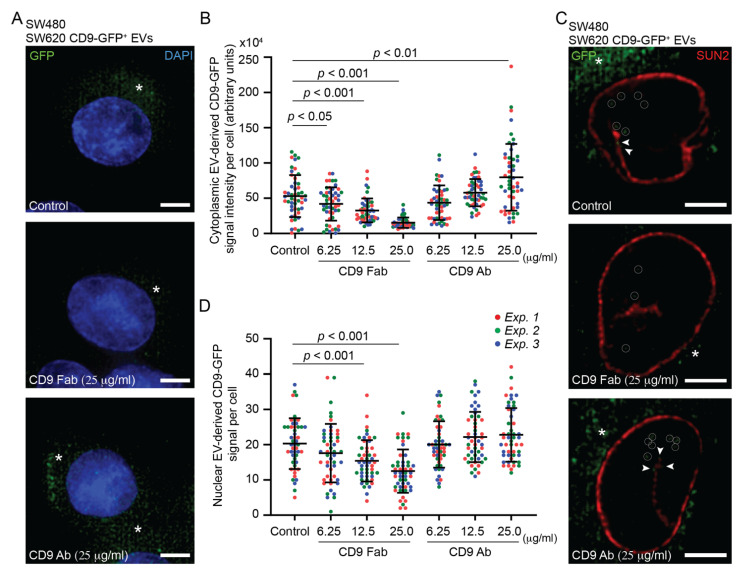
Effects of CD9 Fab and divalent Abs on the internalization of SW620 cell-derived CD9-GFP^+^ EVs into SW480 cells. (**A**–**D**) SW480 cells and fluorescent EVs derived from CD9-GFP^+^ SW620 cells (1 × 10^9^ particles) were individually pre-incubated for 30 min without (control) or with different concentrations of anti-CD9 Fab or divalent Ab as indicated before their co-incubation for 5 h in the absence or presence of Abs (protocol #1). Fixed cells were either stained with DAPI (**A**,**B**) or immunolabeled for SUN2 (**C**,**D**) before observation by CLSM. Note the presence of discrete punctate GFP signals in the cytoplasmic (**A**,**C**) or nucleoplasmic (**C**) compartments (asterisk and circle, respectively). Their intensity (**B**) or amount (**D**) was quantified using serial optical sections through a cell (see Appendix A). Means ± S.D. of individual signals from three independent experiments, as indicated by color coding, are shown (*n* > 15 cells per experiment). *p* values are indicated. Arrowheads indicate CD9-GFP signals in the nuclear envelope invagination (**C**). Scale bars, 5 µm.

**Figure 4 cells-11-02474-f004:**
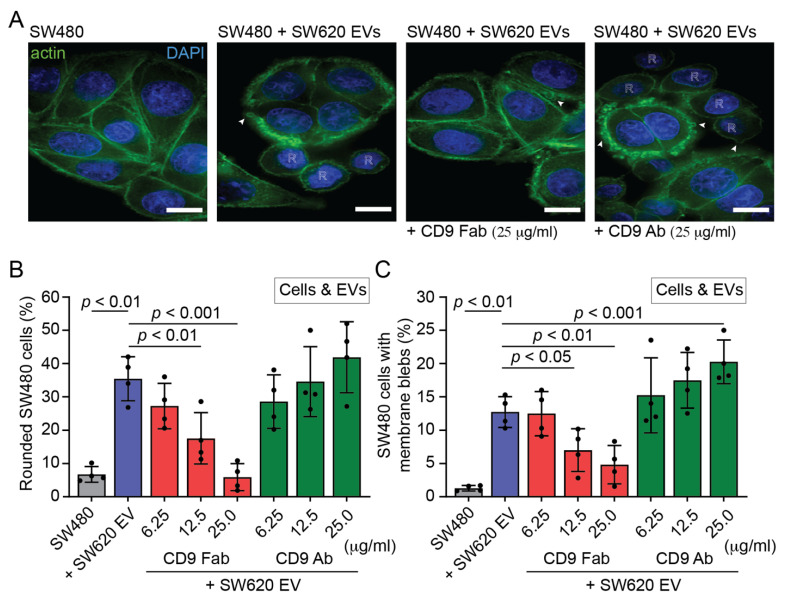
Effects of CD9 Fab and divalent Ab on the pro-metastatic morphological alterations of SW480 cells exposed to SW620 cell-derived EVs. (**A**–**C**) SW480 cells and SW620 cell-derived EVs (1 × 10^9^ particles) were individually pre-incubated for 30 min with different concentrations of anti-CD9 Fab (red) or divalent Ab (green) as indicated before their co-incubation for 5 h in the presence of Abs, as described for protocol #1. As negative and positive controls, cells were not exposed (SW480, grey) or were exposed to EVs in the absence of Ab (+ SW620 EV, blue), respectively. Afterward, fixed cells were stained with DAPI and fluorochrome-conjugated phalloidin to label nuclei and actin filaments, respectively, before observation by CLSM (**A**). Single sections are presented. Rounded cell morphology and membrane blebs induced by EVs are indicated by the letter R and the arrowheads, respectively. The percentage of cells with rounded morphology (**B**) or membrane blebs (**C**) was quantified. Means ± S.D. and individual values for each experiment are shown (*n* = 4). At least 100 cells were evaluated for each experiment. *p* values are indicated. Similar experiments were performed by pre-incubating only EVs or cells with Abs (Appendix A). Scale bars, 10 µm.

**Figure 5 cells-11-02474-f005:**
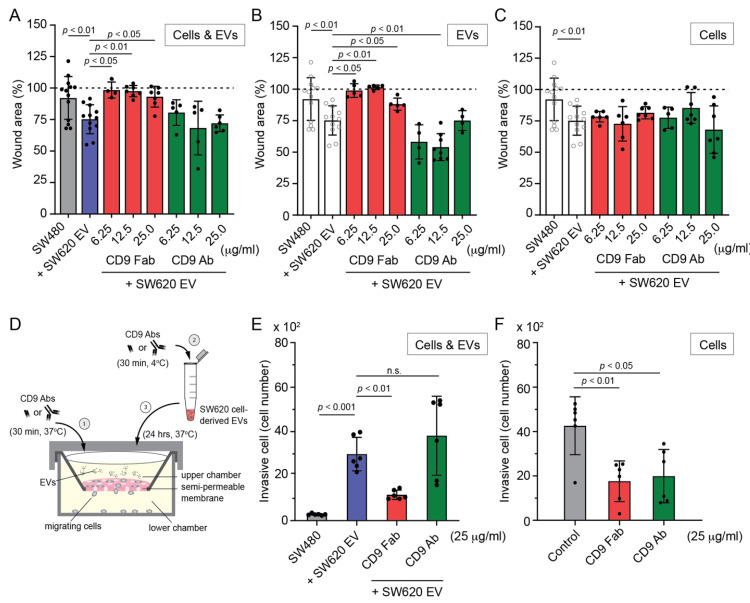
Effects of CD9 Fab and divalent Ab on the migration of SW480 cells exposed to SW620 cell-derived EVs. (**A**) The migration–wound healing assay was performed by introducing a scratch on confluent SW480 cell monolayers cultured on 12-well standard cell culture plates and incubating them for 5 h in the absence (negative control, grey) or presence (positive control, blue) of SW620 cell-derived EVs (1 × 10^9^ particles/mL). Alternatively, after introducing a scratch on the cell monolayer, cells and EVs were individually pre-incubated for 30 min with different concentrations of anti-CD9 Fab (red) or divalent Ab (green) as indicated before their 5 h co-incubation in the presence of Abs (**A**). The percentage of remaining wound areas after 5 h was quantified. Baseline (100%, dashed line) refers to wound area at 0 h. (**B**,**C**) Solely EVs (**B**) or cells (**C**) were pre-incubated for 30 min with Abs prior to co-incubation with cells or EVs for 5 h. Wound area was quantified. Controls (white) are shown for comparison. Means ± S.D. from multiple scratches are shown (*n* = 4–13). (**D**–**F**) The migration–Transwell filter assay was performed using a Transwell chamber as illustrated (**D**), where SW480 (**E**) or SW620 (**F**) cells were added to the upper chamber. Cells and EVs were pre-treated, as described in panel A, using 25 µg/mL CD9 Fab or divalent Ab ((**D**), steps 1 and 2) before 24 h of co-incubation ((**D**), step 3). In the case of SW620 cells, they were not incubated with EVs (**F**). The amount of migrating cells recovered in the lower chamber was then quantified. Each individual value is shown. *p* values are indicated. n.s., not significant.

**Figure 6 cells-11-02474-f006:**
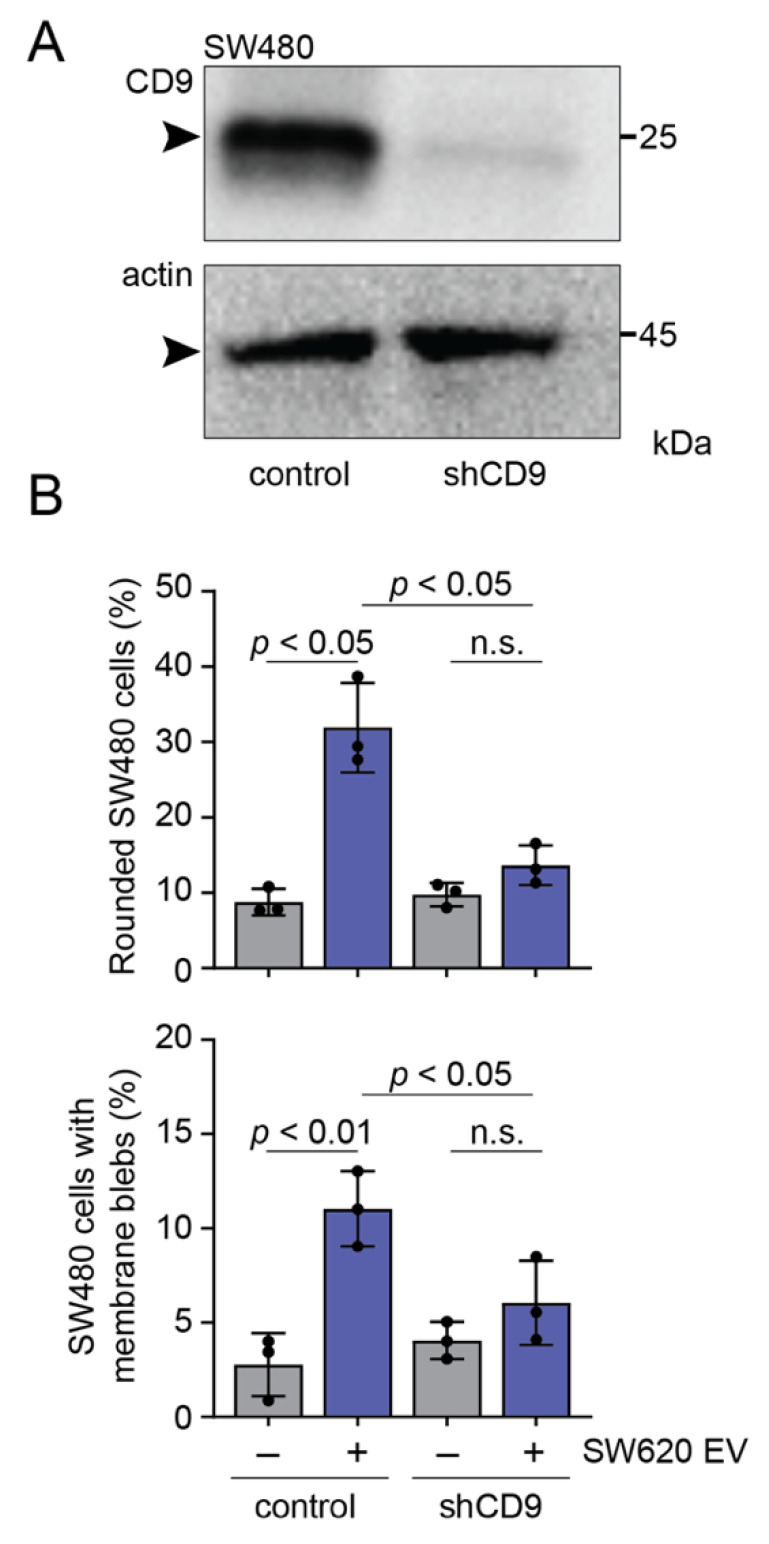
The lack of CD9 in SW480 cells impedes pro-metastatic morphological alterations produced by SW620 cell-derived EVs. (**A**) Parental (control) or CD9-knockdown (shCD9) SW480 cells were analyzed by immunoblotting for CD9 and β-actin. Molecular mass markers (kDa) are indicated. Arrowhead indicates the protein of interest. (**B**) SW480 cells as indicated were incubated with (+) SW620 cell-derived EVs (1 × 10^9^ particles) or without (–) for 5 h. Afterward, fixed cells were stained with DAPI and fluorochrome-conjugated phalloidin to label nuclei and actin filaments, respectively, before observation by CLSM. The percentage of cells with rounded morphology (top panel) or membrane blebs (bottom panel) was quantified. Means ± S.D. and individual values for each experiment are shown (*n* = 3). At least 100 cells were evaluated for each experiment. *p* values are indicated. n.s., not significant.

## Data Availability

The raw data supporting the conclusions of this article will be made available by the authors, without undue reservation.

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
