# Peer review of "Anti-Human CD9 Fab Fragment Antibody Blocks the Extracellular Vesicle-Mediated Increase in Malignancy of Colon Cancer Cells"

_cells, 2022, doi:10.3390/cells11162474_

Round 1

Reviewer 1 Report

I would like to congratulate the Authors the submitted manuscript. The presented data on targeting EVs in colon cancer are of great importance in context of future clinical applications especially. Several comments and suggestions appeared when reviewing the manuscript that could slightly improve already really good manuscript. These comments are pointed out below.

Introduction.

That part is well-written and provides comprehensive but still succinct introduction to the rest of the manuscript. The language used is perfect and does not require verification. The authors provide clear and specific aim of the study.

Methods.

1)    The protocols implemented in the project are perfectly designed, however, I just wondered whether the authors had chance to test how CD9 Ab / CD9 Fab can affect the observed results? Shortly, will tested reagents be only effective when applied before presence of SW620 EVs or they could also provide beneficial effects when EVs already started to induce certain effects?

Results.

1)    Apart from analysis of CD9+ SW620 EVs internalization, did authors analyze changes in CD9 on SW480? I see that SW480 were supposed to be CD9-deficient in accordance with the protocol but was it a complete lack of CD9? As demonstrated, CD9 was normally present on these cells so how, hypothetically, the presented effects would differ in conditions with CD9+ SW480 cells?

2)    Could the authors provide more extensive description of calculations made for the results presented in figure S2? Please correct me if I am wrong in my interpretation. In triple staining in graph B, it seems that low frequency results from including only EVs that have all three markers present. However, in graph A such combination increased frequency, thus, suggesting that sum of all EVs with CD9, CD63 or CD81 was used to achieve high frequency?

3)    Regarding graphs C and D in figure S3, would it be possible for the authors to additionally include indication of statistical differences compared to ‘Controls’ with simple asterisks above each of the columns?

4)    Sample representative photos for data presented within figure S4 would be beneficial for the manuscript. These corresponds to the figure 5 so maybe additional representative photos of most significant results could also be included within the manuscript?

5)    I think that it would be essential to use additional microscopic presentation of the results described in the figure 6.

Discussion.

1)    The authors studied and discussed the role of EVs in reference to intercellular communication of metastatic (SW620) and primary (SW480) colon cancer cells. How other cells also releasing CD9-positive EVs, already mentioned within introduction, can affect observed results?

2)    In the subsection 3.5 of the results, the authors suggest other factors could be responsible for changes in SW480 morphology apart from tested CD9. Could that topic be also shortly discussed within the discussion section?

3)    I think that the summary provided within the last paragraph of the discussion section could be extended. The authors did not only demonstrate significant effects of CD9 Fab implementation on the colon cancer cells function in vitro. In addition, some essential CD9- and EVs-related mechanism were shown and these should also be mentioned as the significant outcomes of the study.

4)    Undoubtedly, the manuscript will benefit from providing graphical abstract that would succinctly summarize the project background and most significant results obtained. 

Author Response

Reviewer 1
Comments and Suggestions For Authors”
I would like to congratulate the Authors the submitted manuscript. The presented data on targeting EVs in colon cancer are of great importance in context of future clinical applications especially. Several comments and suggestions appeared when reviewing the manuscript that could slightly improve already really good manuscript. These comments are pointed out below.

Author Response
We appreciate the reviewer's very positive evaluation of our work and thank her/him for valuable and constructive suggestions.

Reviewer 1’s comment
Introduction.
That part is well-written and provides comprehensive but still succinct introduction to the rest of the manuscript. The language used is perfect and does not require verification. The authors provide clear and specific aim of the study.

Author Response
Again, we appreciate her/his comments about the Introduction. As suggested by the second reviewer, we provided additional information about the content of EVs.

Reviewer 1’s comment
Methods.
1)    The protocols implemented in the project are perfectly designed, however, I just wondered whether the authors had chance to test how CD9 Ab / CD9 Fab can affect the observed results? Shortly, will tested reagents be only effective when applied before presence of SW620 EVs or they could also provide beneficial effects when EVs already started to induce certain effects?
Author Response
That's a good thought. Upon endocytosis, cancer cell-derived EVs, including a fraction of their cargo that reaches the nucleoplasm of recipient cells could already induce its cellular transformation. Addition of an anti-CD9 Fab would probably not reverse the ongoing effect. The data obtaining with protocol #3’, where CD9 Fab is removed prior the addition of EVs, is consistent with this notion (Supplementary Fig. S5C). However, such a strategy could interfere with reciprocal communication between EV-primed recipient cells and donor cells in primary and secondary cancer sites, and thus inhibit cancer cell growth. This idea could be studied in three-dimensional cultures using cancer spheroids and stromal cells, e.g., fibroblast mesenchymal stromal cells.

Reviewer 1’s comment
Results.
1)    Apart from analysis of CD9+ SW620 EVs internalization, did authors analyze changes in CD9 on SW480? I see that SW480 were supposed to be CD9-deficient in accordance with the protocol but was it a complete lack of CD9? As demonstrated, CD9 was normally present on these cells so how, hypothetically, the presented effects would differ in conditions with CD9+ SW480 cells?
Author Response
By applying lentiviral-mediated short hairpin RNA, more than 90% of CD9 is silenced in SW480 cells, as demonstrated by immunoblotting (see section 3.5 and Fig. 6A). This significant reduction did alter the general mesenchymal-like morphology of SW480 cells, but impeded their cellular transformation mediated by EVs derived from SW620 cells. The latter data suggest that cell surface CD9 would mediate the initial binding of CD9+ EVs prior to their internalization. However, it remains to determine whether CD9 would directly participate in their internalization as described for another tetraspanin protein, Tspan8, which interacts with intersectin-2, known to promote endocytosis (Rana et al., 2011, PMID:20937409). In light of this, we have slightly modified the corresponding text in section 3.5.
Regarding the general morphology of SW480 cells in the absence of CD9, other related proteins in the tetraspanin web might contribute to the general organization of the plasma membrane and adhesion with the underlying substrate. At this stage, we have not studied in these cells whether other tetraspanin molecules are upregulated in the absence of CD9. Nevertheless, we have previously shown that CD9 knockdown upregulates CD81 expression in breast cancer cells (Rappa et al., 2015, PMID:25762645). We added this information in the revised Discussion.

Reviewer 1’s comment
2)    Could the authors provide more extensive description of calculations made for the results presented in figure S2? Please correct me if I am wrong in my interpretation. In triple staining in graph B, it seems that low frequency results from including only EVs that have all three markers present. However, in graph A such combination increased frequency, thus, suggesting that sum of all EVs with CD9, CD63 or CD81 was used to achieve high frequency?
Author Response
In both panels, all small (A) and large (B) EVs were triple-labeled with the three tetraspanin proteins. The results show that the main differences between these samples are that triple-positive EVs are very abundant in the small EV population, whereas those that are only positive for CD9 are enriched in the large EV population. The latter suggest that they may originate from plasma membrane and are therefore considered as microvesicles/ectosomes, while triple-positive small EVs might represent exosomes, i.e. those released after the fusion of multivesicular body with the plasma membrane.

Reviewer 1’s comment
3)    Regarding graphs C and D in figure S3, would it be possible for the authors to additionally include indication of statistical differences compared to ‘Controls’ with simple asterisks above each of the columns?
Author Response
Done.

Reviewer 1’s comment
4)    Sample representative photos for data presented within figure S4 would be beneficial for the manuscript. These corresponds to the figure 5 so maybe additional representative photos of most significant results could also be included within the manuscript?

Author Response
We added representative images of the different conditions used as requested.

Reviewer 1’s comment
5)    I think that it would be essential to use additional microscopic presentation of the results described in the figure 6.

Author Response
The data presented in Figure 6 are similar to those presented in Figures 3 and S4. It seems redundant to show the same thing and for this reason we have presented only the quantification.

Reviewer 1’s comment
Discussion.
1)    The authors studied and discussed the role of EVs in reference to intercellular communication of metastatic (SW620) and primary (SW480) colon cancer cells. How other cells also releasing CD9-positive EVs, already mentioned within introduction, can affect observed results?

Author Response
In a given specific microenvironment, EV-mediated intercellular communication is a complex system in which EVs released from different cell types can enable simultaneous exchanges between them. Of course, cells primed by EVs may respond differently by altering the content of the EVs. This last point is particularly true in the case of cancer where transformed cells secrete a large amount of EVs in order to modify their microenvironment or niche that would favor their expansion and dissemination. In our context, the use of CD9 Fab is aimed at intercepting EVs released by CD9+ cancer cells, but other healthy CD9+ cells can also release EVs, albeit to a lesser extent, that can technically neutralize CD9 Fab. For these reasons, the next step will be to analyze this protocol in animal models to determine the amount of monovalent antibody and the frequency of injection that would block cancer cell-derived EV-mediated intercellular communication. We added this concept at the end of the Discussion.

Reviewer 1’s comment
2)    In the subsection 3.5 of the results, the authors suggest other factors could be responsible for changes in SW480 morphology apart from tested CD9. Could that topic be also shortly discussed within the discussion section?

Author Response
As discussed above (see Results, point #1), other proteins involved in the tetraspanin web could contribute to the overall organization of the plasma membrane and thus to their adhesion allowing the maintenance of their mesenchymal-like morphology in the absence of CD9. Although, we did not investigate this further, upregulation of CD81 (or other tetraspanins) could explain the lack of a major phenotype in CD9-deficient SW480 cells (see above). We present some thoughts on this in the revised Discussion.

Reviewer 1’s comment
3)    I think that the summary provided within the last paragraph of the discussion section could be extended. The authors did not only demonstrate significant effects of CD9 Fab implementation on the colon cancer cells function in vitro. In addition, some essential CD9- and EVs-related mechanism were shown and these should also be mentioned as the significant outcomes of the study.

Author Response
Following the reviewer's suggestion, we have expanded our final summary to cover more extensively our results.

Reviewer 1’s comment
4) Undoubtedly, the manuscript will benefit from providing graphical abstract that would succinctly summarize the project background and most significant results obtained.

Author Response
We agree with the reviewer and now provide a Graphical Abstract

Reviewer 2 Report

The manuscript by Santos and colleagues describes, in vitro, a new potential approach to improve anti-cancer therapeutic strategies by interfering with the intercellular communication mediated by extracellular vesicles (EVs). In its current form the manuscript is clear and solid. It needs only minor revision before it is suitable for publication in Cells.

1.       The introduction (lines 49-51) and discussion could be improved with: a more detailed description of the content of the EVs in the SW620 specific model; and considerations on the possible mediators of cell transformation and migration in the EVs content.

2.       In Materials and Methods (line 176), the temperature of low-speed centrifugations is not specified.

3.       In Materials and Methods (line 216), replace “in 4°C” with “at 4°C”.

4.       In Materials and Methods (line 350), check the volume of the medium added to the upper chamber.

Author Response

Reviewer 2
Comments and Suggestions For Authors”
The manuscript by Santos and colleagues describes, in vitro, a new potential approach to improve anti-cancer therapeutic strategies by interfering with the intercellular communication mediated by extracellular vesicles (EVs). In its current form the manuscript is clear and solid. It needs only minor revision before it is suitable for publication in Cells.

Author Response
We are pleased to read that the second reviewer also appreciated our works, and we thank her/him for the suggestion and carefully reading of our manuscript.

Reviewer 2’s comment
1. The introduction (lines 49-51) and discussion could be improved with: a more detailed description of the content of the EVs in the SW620 specific model; and considerations on the possible mediators of cell transformation and migration in the EVs content.

Author Response
First, we added some general information in the introduction about the content of EVs, including the type of nucleic acids they can carry. It is also mentioned that EVs can reprogram the fate of recipient cells and stimulate their migration (see the end of the first paragraph). Second, we have added specific information in the revised Discussion about the content of SW620 cell-derived EVs, including RhoA interactors that activate the RhoA/ROCK pathway, which is known to induce amoeboid cell migration. 

Reviewer 2’s comment
2. In Materials and Methods (line 176), the temperature of low-speed centrifugations is not specified.

Author Response
Thank you for noticing this. We have added the relevant information in the revised manuscript.

Reviewer 2’s comment
3. In Materials and Methods (line 216), replace “in 4°C” with “at 4°C”.

Author Response
Done.

Reviewer 2’s comment
4. In Materials and Methods (line 350), check the volume of the medium added to the upper chamber.

Author Response
Sorry for our mistake. We have corrected it.